# Transcriptome Changes in Glioma Cells Cultivated under Conditions of Neurosphere Formation

**DOI:** 10.3390/cells11193106

**Published:** 2022-10-02

**Authors:** Natalia S. Vasileva, Elena V. Kuligina, Maya A. Dymova, Yulya I. Savinovskaya, Nikita D. Zinchenko, Alisa B. Ageenko, Sergey V. Mishinov, Anton S. Dome, Grigory A. Stepanov, Vladimir A. Richter, Dmitry V. Semenov

**Affiliations:** 1Institute of Chemical Biology and Fundamental Medicine, Siberian Branch, Russian Academy of Sciences, Lavrentyev Avenue 8, Novosibirsk 630090, Russia; 2Novosibirsk Research Institute of Traumatology and Orthopedics n.a. Ya.L. Tsivyan, Department of Neurosurgery, Frunze Street 17, Novosibirsk 630091, Russia

**Keywords:** glioma, glioblastoma, cancer stem cells, neurospheres, epithelial to mesenchymal transition, pro-neural to mesenchymal transition

## Abstract

Glioma is the most common and heterogeneous primary brain tumor. The development of a new relevant preclinical models is necessary. As research moves from cultures of adherent gliomas to a more relevant model, neurospheres, it is necessary to understand the changes that cells undergo at the transcriptome level. In the present work, we used three patient-derived gliomas and two immortalized glioblastomas, while their cultivation was carried out under adherent culture and neurosphere (NS) conditions. When comparing the transcriptomes of monolayer (ML) and NS cell cultures, we used Enrichr genes sets enrichment analysis to describe transcription factors (TFs) and the pathways involved in the formation of glioma NS. It was observed that NS formation is accompanied by the activation of five common gliomas of TFs, SOX2, UBTF, NFE2L2, TCF3 and STAT3. The sets of transcripts controlled by TFs MYC and MAX were suppressed in NS. Upregulated genes are involved in the processes of the epithelial–mesenchymal transition, cancer stemness, invasion and migration of glioma cells. However, MYC/MAX-dependent downregulated genes are involved in translation, focal adhesion and apical junction. Furthermore, we found three EGFR and FGFR signaling feedback regulators common to all analyzed gliomas—SPRY4, ERRFI1, and RAB31—which can be used for creating new therapeutic strategies of suppressing the invasion and progression of gliomas.

## 1. Introduction

Among all registered malignant neoplasms, brain and CNS tumors account for 3.5%, whereas gliomas represent 81% of malignant brain and CNS tumors [1]. The most common histological form of glioma is glioblastoma (approximately 45% of all gliomas) [2]. In general, glioblastoma (GBM) is a rare tumor with a global incidence of less than 10 per 100,000 people, but it remains an incurable disease and one of the most aggressive tumors, characterized by an acute course of disease and poor prognosis—a 5-year survival rate of less than 5% with full standard treatment, and a median survival of only 15 months [3]. The poor prognosis of this cancer is due to aggressive diffuse infiltrative growth, drug resistance, tumor heterogeneity, immune evasion, and obstacles to drug delivery, such as the presence of the blood–brain barrier and the blood–brain tumor barrier [4]. The standard treatment of GBM, which today consists of the maximum surgical resection of the tumor followed by radio- and/or chemotherapy, is not effective; therefore, the development of new approaches for diagnostics and therapy remains an urgent task.

In the development of anticancer drugs, various cellular and animal models are used in order to evaluate the cytotoxic effect of the drug in vitro and its antitumor efficacy in vivo. A significant part of the research on the effectiveness of anticancer drugs is carried out using immortalized cell cultures. However, cells of immortalized lines undergo differentiation during long-term cultivation, and the cell culture no longer reflects the cellular heterogeneity, genetic and morphological features that are characteristic of a malignant tumor, including glioblastoma. One solution to this substantial problem is to use cancer cell cultures derived from patient biopsy as an in vitro model [5].

The cultivation of primary tumor cell cultures in media-containing serum leads to the differentiation of tumor cells and the loss of their tumorigenic potential [6]. On the other hand, the cultivation of primary cultures of malignant glioma in a medium without serum, but with the addition of growth factors and nutritional supplements, allows us to form NS [7], which is enriched in cells with characteristic features of tumor stem cells: the potential for unlimited self-renewal and the ability to readily differentiate to into neurons and glial cells. Moreover, these cells demonstrate gene expression profiles and biological behavior that maximally preserved both the phenotype and genotype of the primary tumor and the molecular and phenotypic features of cancer stem cells (CSCs) [8]. They closely mimic their parental primary tumors, in contrast to tumor cells grown under standard culture conditions. In fact, standard culture conditions cannot enrich CSCs, which ultimately leads to an increase in the population of cells that remotely resemble the original tumors.

In 1992, Brent A. Reynolds and Samuel Weiss were the first to demonstrate the ability of neural stem cells to form NS in the presence of the epidermal growth factor (EGF) and a non-adherent substrate [9]. Yuan X. et al. showed that the GBM spheres share many characteristics of stem cells, including self-renewal ability and multipotent differentiation, which can produce daughter cells of all phenotypes present in the GBM. After in vivo implantation, only the isolated tumor stem cells were able to form tumors that contained both neurons and glial cells [10]. Therefore, the analysis of glioma cells in neurospheres quickly became the method of choice, and since has become a valuable tool for isolating and understanding the biology of embryonic and CSCs because the latter are a huge obstacle to the treatment of glioblastoma, due to their drug resistance, radiation resistance, unlimited potential for self-renewal and participation in immune evasion [11,12,13]. Moreover, patient-derived glioma cells cultivated in the form of NS retain the ability to form tumors upon xenotransplantation in immunodeficient animals and thus may be a promising and relevant model for studying the mechanisms of the tumor development, the key glioblastoma signaling pathways, the features of clinical outcome and the response to anticancer drugs [14,15,16,17,18].

For the formation of neurospheres, GBM cells are cultivated in a medium with the addition of growth factors, such as EGF and the fibroblast growth factor (bFGF), thereby artificially activating small GTP-ase—RAS protein. This protein triggers many important downstream signaling pathways resulting in the growth, adhesion, cytoskeletal integrity, differentiation, survival and migration of cells [19]. First, it is the mitogen-activated protein kinase (MAPK) pathway that activates transcription factors such as MYC, FOS, ETS, and JUN, thus promoting cell cycle entry, angiogenesis, and survival by supporting cancer cell proliferation [20]. RAS induces the phosphatidylinositol 3-kinase (PI3K)/protein kinase B (Akt) pathway, the increased activity of which leads to apoptosis evasion and is associated with tumor progression and drug resistance. Moreover, RAS through TIAM1 regulates cytoskeletal organization and cell migration [21]. The RAS/MAPK signaling pathway is believed to induce epithelial–mesenchymal transition (EMT). It stimulates the nuclear expression of EMT-inducing transcription factors and reprograms the expression of genes that are involved in intercellular adhesion, cytoskeletal positioning, invasion and migration [22]. Specifically, for gliomas, the term pro-neural to mesenchymal transition (PMT) is used [23]. Studies of key molecular genetic markers associated with PMT began relatively recently; the activation of such classic transcription factors for oncogenesis as STAT3 and NF-κB has been identified [24].

In this work, a NGS transcriptomic analysis followed by the comparative bioinformatic analysis of glioma cell RNA patterns in the adherent state and in the state of neurospheres was performed. SOX2, UBTF, NFE2L2, TCF3 and STAT3 were underlined as common transcriptional factors, which are responsible for the upregulation of genes involved in processes of the epithelial–mesenchymal transition, cancer stemness, invasion and migration of GBM. We also detected that the upregulation of SPRY4, ERRFI1 and RAB31 provides a condition for feedback regulations of FGF and EGF signaling as well as can be applied in glioma therapy.

## 2. Materials and Methods

### 2.1. Cell Lines

Human U87 MG and U343 MG cell lines were obtained from the cell culture collection of the Institute of Molecular and Cellular Biology of the SB RAS (Novosibirsk, Russia). The cells were cultivated in Minimum Essential Medium α (MEM α; Sigma-Aldrich, Saint Louis, MO, USA) with 10% FBS (Gibco BRL Co., Gaithersburg, MD, USA), 2 mM L-glutamine (Sigma-Aldrich, USA), 250 mg/mL amphotericin B, and 100 U/mL penicillin/streptomycin (Gibco BRL Co., Gaithersburg, MD, USA) at 37 °C in a humidified atmosphere containing 5% CO_2_.

### 2.2. Patient-Derived Cell Cultures

Cancer tissue samples were obtained with informed consent from patients at the Novosibirsk Research Institute of Traumatology and Orthopedics n.a. Ya.L. Tsivyan (Novosibirsk, Russia). The study was approved by the Committee on the Ethics of Novosibirsk Research Institute of Traumatology and Orthopedics n.a. Ya.L. Tsivyan (protocol number № 050/17 68 of 11 September 2017).

All samples were collected from treatment-naïve patients. According to the transcriptome data, the analyzed glioma cultures do not have mutations in the coding regions of the IDH1 and IDH2 mRNAs.

Glioma tissue specimens were mechanically dissociated in Iscove’s modified Dulbecco’s media (IMDM, Sigma-Aldrich, Saint Louis, MO, USA). Specimens dissociated into single cells were washed with a 10× excess volume of phosphate-buffered saline (PBS), and separated cells were collected through centrifugation at 300× *g*. Cells were plated in IMDM medium with 10% FBS, 2 mM L-glutamine, 100 U/mL penicillin, 100 μg/mL streptomycin, and 250 mg/mL amphotericin B for cell adhesion. At the next passages, cells were cultured in complete IMDM medium supplemented with Mito + Serum Extender (BD Biosciences—Discovery Labware, San Jose, CA, USA), 2 mM L-glutamine, 100 U/mL penicillin, 100 μg/mL streptomycin, and 250 mg/mL amphotericin B, and were cultivated in 6-well plates at 37 °C in a humidified atmosphere containing 5% CO_2_. When 70–80% confluence was reached, cells were harvested using TripLE Express (Gibco BRL Co., Gaithersburg, MD, USA) and subcultured for further experiments. The cell cultures tested negative for mycoplasma contamination.

### 2.3. Cell Culture for Neurosphere Formation

For neurosphere formation, cells were cultured in Dulbecco’s Modified Eagle Medium: Nutrient Mixture F-12 (DMEM: F12, Sigma-Aldrich, Saint Louis, MO, USA) supplemented with B-27 and N-2 Supplements, 20 ng/mL bFGF (Gibco BRL Co., Gaithersburg, MD, USA) and 20 ng/mL EGF (Sigma-Aldrich, Saint Louis, MO, USA) in non-treated cell culture dishes (Eppendorf, Germany) at 37 °C in a humidified atmosphere containing 5% CO_2_. Phase-contrast microscopy was performed using the Nicon Eclipse Ti-S microscope (Nikon, Japan). U87 MG and U343 MG were collected at 3 passages. BR1 and BR3 were collected at 4 passages, and MG1 was collected at 8 passages.

### 2.4. RNA Isolation

Total RNA was extracted from cells with an RNA extraction kit (LRU-100-50, Biolabmix, Russia) following the manufacturer’s protocol. The RNA concentration was assessed using the Qubit 2 fluorometer (Thermo Fisher Scientific, Waltham, MA, USA) with the Qubit RNA HS Assay Kit (Thermo Fisher Scientific, USA). The quality of total RNA expressed as the RNA Integrity Number (RIN) was determined using the Bioanalyzer 2100 instrument (Agilent, Santa Clara, CA, USA) with an Agilent RNA Pico 6000 Kit (Agilent, Santa Clara, CA, USA). A threshold RIN reading of greater than 8.0 was taken as the cut-off point for the transition to the stage of library preparation.

### 2.5. RNA Sequencing

The construction of Illumina cDNA libraries was performed according to a standard protocol using a NEBNext Ultra II Directional RNA library preparation kit (New England Biolabs, Ipswich, UK) and NEBNext mRNA Magnetic Isolation Module (New England Biolabs, UK), as well as massive parallel sequencing on a NextSeq Illumina 1500 platform, at the Institute of Fundamental Medicine and Biology, Kazan Federal University (Kazan, Russia). For the isolation of mRNA, and the fragmentation and priming procedure, 1 μg of the total RNA was used. A NextSeq 500/550 High Output v2.5 Kit (100-nucleotide single-end reads) (Illumina, San Diego, CA, USA) was used. For the prepared sequencing libraries, fragment size distribution was analyzed using a Bioanalyzer 2100 instrument (Agilent, USA) with an Agilent High Sensitivity DNA Kit (Agilent, USA) and quantified using the Qubit 2.0 Fluorometer (Invitrogen, Waltham, MA, USA) with the Qubit dsDNA HS Assay Kit (Thermo Fisher Scientific, USA). Fragment sizes ranged between 250 bp and 700 bp, with a clear peak at 300 bp.

### 2.6. Transcriptome Analysis

Raw sequencing reads (100-nucleotide single-end reads) were subjected to Illumina adapter removal using Trimmomatic [25]. Adapter trimmed sequencing reads were filtered with Bowtie2 [26] using a reference containing sequences of human: rRNAs (RefSeq); tRNAs; snRNA; SINE-, LINE-, and DNA-repeat consensus sequences (RepBase [27]); low-complexity simple repeats, as well as mitochondrial DNA (NC_012920.1). Filtered reads were mapped to a human genome (GRCh37/hg19) with STAR 2.7.1a [28] using RefGene human genome annotation (https://hgdownload.cse.ucsc.edu/goldenPath/hg19/database/ accessed on 5 June 2021). Aligned reads were quantified using QoRTs v1.3.6 [29]. Differential gene expression analysis was performed with DESeq2 1.36.0 [30], R version 4.1.3 and Bioconductor 3.14. The results of differential gene expression analysis, lists of up/downregulated genes, were analyzed with Enrichr using R interface [31]. We also used GSEA MsigDB for the independent analysis of gene sets [32,33].

### 2.7. Real Time RT-PCR Analysis of RNA

To confirm the RNAseq results with qRT-PCR, we randomly selected 8 mRNAs: CXCL1, ERRFI1, NFKBIA, NRP2, PDGFRA, SOX2, TRIB2, and ZEB1. Total cellular RNA was isolated using an RNA extraction kit (LRU-100-50, Biolabmix, Russia) with additional Dnase I and Rnase-free (Thermo Fisher Scientific, USA) digestion according to the manufacturer’s protocol. The forward and reverse primers were synthesized in ICBFM SB RAS, Russia (Appendix A). Real-time PCR was performed on a CFX96 Touch Real-Time PCR Detection System (Bio-Rad, Hercules, CA, USA), using the reagent kit BioMaster RT-PCR SYBR Blue (2×) (Biolabmix, Novosibirsk, Russia). The RT-PCR conditions included the synthesis of cDNA at 45 °C for 30 min, initial activation at 95 °C for 5 min, 40 cycles with denaturation at 95 °C for 10 s, annealing at 61.5 °C for 20 s and extension step at 72 °C for 30 s selected for primers GAPDH, SOX2, ERRFI, TRIB2, and NRP2, or annealing at 58 °C for 10 s and an extension step at 72 °C for 20 s selected for primers GAPDH, NFKBIA, CXCL1, PDGFRA, and ZEB1. The melting curves were analyzed to ensure the specificity of the products. Each sample was analyzed in triplicate. The levels of mRNA were represented as relative values normalized to the level of GAPDH. To confirm the amplification of targeted gene fragments, PCR products were separated through electrophoresis in 1.5% agarose gel, stained with ethidium bromide and documented with the Gel Doc XR System (Bio-Rad, Hercules, CA, USA).

### 2.8. Statistical Analysis

Data are presented as the mean ± SD. The number of replicates for each experiment is stated in the figure legends. Statistical differences between the 2 groups were evaluated using a 2-tailed t-test; *p* < 0.05 was considered to be statistically significant.

## 3. Results

### 3.1. Neurosphere Formation from Primary Brain Tumors and Immortalized Cell Lines

In order to obtain patient-derived glioma cell cultures, we used three solid primary brain tumors (Table 1), which were acutely dissociated into individual cells. We used culture conditions that favored stem cell growth, developed for the isolation of neural stem cells in the form of neurospheres [7,34,35]. We also cultivated cells of immortalized lines U87 MG and U353 MG in the same conditions for neurosphere formation (Table 1).

Both patient-derived and immortalized glioma cells formed neurospheres (Figure 1A). Neurospheres obtained from immortalized cells formed faster, reaching a size of approximately 150 μm within 3–4 days. The efficiency of neurosphere formation by cells obtained from patients varied significantly for different cell cultures. Thus, MG1 cells formed spheres of 150 μm in size by 8–10 days of cultivation, while BR2 cells formed neurospheres within 3–5 days.

### 3.2. The Formation of Neurospheres Occurs in a Common Way in Patient-Derived Glioma Cell Cultures and Immortalized Cell Lines

We performed transcriptome analysis based on Illumina 1500 NGS platform and obtained from ~1.9 × 10^7^ to ~4.6 × 10^7^ experimental reads for each of gliomas cell cultures in adherent and neurosphere (Table 1).

To create a general description of transcriptome changes under the condition of NS formation, we started with the principal component analysis and hierarchical clustering (HC) of RNA sequencing data. The PCA shows that the formalized RNA expression data of immortalized cell lines U87 MG and U343 MG form distinct non-overlapping areas of points with that of the patient-derived glioma cultures BR1, BR2 and MG1. Patient-derived cells tend to form closely related and even overlapping areas in PCA graphs (Figure 1B). The HC, in conformity with PCA, shows that the transcriptomes of U87 a/n as well as U343 a/n cells form separate branches of the tree when compared to patient-derived cultures (Figure 2). In HC, clade included the adherent cultures BR1a, BR2a and MG1a consisting of separate branches, which are clearly distinguished from the corresponding neurospheres (Figure 1 and Figure 2).

This indicates that transcriptome changes in the process of glioma NS formation are determined to a greater extent by the initial cell-specific context of gene expression, but strongly modulated by the conditions of cultivation in the presence of bFGF and EGF. In light of this, the common trend of PC1:PC2 coordinate changes is observed for all analyzed gliomas (Figure 1B). Thus, the formation of NS has common transcriptional features and, possibly, common gene patterns for all analyzed gliomas.

### 3.3. Common Gene Expression Changes in Both Patient-Derived and Immortalized Glioma Cell Cultures

To search for transcripts with similar trends in expression changes in the conditions of glioma NS formation, we used a direct approach, including the comparison of transcriptomes in specific monolayer/neurosphere (MN/NS) pairs separately and the determination of common overlapping transcripts for all analyzed pairs (Figure 3A).

We found 203 synchronously activated and, separately, 154 synchronously repressed overlapping transcripts in the gene expression sets of individual MN/NS pairs (Figure 3, Table 2).

To describe the common features of glioma cell NS formation, we used Enrichr [31]. For the overall description of TFs, controlling gene expression changes, we explored the Enrichr library “ENCODE and ChEA Consensus TFs from ChIP-X”. From the data in Table 3, it can be seen that in the sets of all upregulated transcripts, statistically significant increases were observed in those controlled by the transcription factors SOX2, UBTF, and NFE2L2.

SOX2, UBTF and NFE2L2 are among the 10 most significant factors responsible for increased expression both in particular MN/NS pairs and in the group of common overlapping genes. The downregulation of the activity of transcription factors MYC and MAX is observed in the Enrichr data for both specific MN/NS pairs and the corresponding set of overlapping genes (Table 3 and Appendix A).

#### 3.3.1. SOX2-Dependent Gene Activation under Conditions of Neurosphere Formation by Glioma Cells

It is known that the overexpression of transcription factor SOX2 has been found in different human cancers, including glioma. SOX2 regulates cell processes by activating or repressing target genes via binding its promoters. Traditionally, in the case of malignant neoplasms, the activation of SOX2 is associated with the development and maintenance of the stemness of tumor cells, increased cell proliferation, the activation of Wnt/β-catenin signaling, JAK/STAT3 signaling, apoptosis evasion, EMT promotion, invasion and metastasis. Additionally, different studies have shown the participation of SOX2 conferring drug resistance [36,37].

We found that the basic level of SOX2 mRNA significantly differs in glioma cells. The baseline level of SOX2 mRNA in immortalized cells is higher than in cells of patient-derived cultures. In light of this, for four out of the five analyzed cultures (except for MG1), an increase in the SOX2 mRNA level in NS cell cultures was detected (Figure 4).

Thus, changes in the relative level of SOX2 mRNA were not unidirectional for all analyzed cell cultures under conditions of NS formation. At the same time, there was a significant enrichment in SOX2-dependent transcripts in the lists of activated genes of individual cell cultures, as well as in the overlapping list of genes (“SOX2 CHEA” in Table 3 and Appendix A, Figure 5). It can be proposed that the observed activation of the SOX2 TF is determined to a greater extent through the post-translational modifications of the SOX2 protein and/or its interaction with other factors, rather than through its level of mRNA in glioma cells.

Among the SOX2-dependent genes, the following should be highlighted: ITGAV, BMP2, SPRY4, NRP2, and SEMA3A. The ITGAV (Integrin αvβ3) plays a key role in FGF/FGFR signaling [38]. BMP2 (Bone Morphogenetic Protein 2) acts as a ligand for TGF-beta receptors that activate SMAD family transcription factors [39]. SEMA3A (Semaphorin 3A) is known to promote the invasion and migration of glioma cells, while NRP2 (Neuropilin-2) regulates the migratory ability of glioma cells in response to SEMA3A [40]. Importantly, SPRY4 is a factor that suppresses FGF/FGFR signaling by interacting with serine/threonine-protein kinase RAF1 and inhibiting its activity [41]. Therefore, the SOX2-dependent activation of SPRY4 provides feedback regulation of FGF/FGFR signaling in conditions of NS formation.

Thus, the activation of SOX2-dependent transcription creates a unified basis for the processes of intercellular interaction at the level of the FGF and TGF-beta/SMAD signaling pathways during the formation of NS by glioma cells. In addition, the activation of gene transcription whose products mediate the invasion and migration of glioma cells may indicate the triggering of the pro-neural to mesenchymal transition (Figure 6).

#### 3.3.2. Activation of UBTF-Dependent Genes

UBTF (Upstream Binding Transcription Factor, UBF1), which is known as a key component of the Pol I pre-initiation complex, mediates the recruitment of RNA polymerase I to rDNA promoter regions. In light of this, UBTF is also involved in the modulation of RNA polymerase II transcription [42].

In our data, the level of UBTF mRNA did not undergo unidirectional changes during the transition from MN to NS (Figure 4). Therefore, it can be proposed that the observed increase in the expression of UBTF-dependent genes (Figure 4 and Figure 5, Table 3 and Appendix A) is associated with the post-translational activation of the protein and/or its interaction with other factors.

The list of UBTF-upregulated genes includes ITGAV, SPRY4 (see above), MAML3, SMAD5, TRIB2 and ZEB1 (Appendix A). MAML3 acts as a transcriptional coactivator for Notch proteins and Notch signaling in the nucleus [43]. SMAD5 is one of the key participants in the BMP signaling pathway, functioning as a transcriptional modulator activated by BMP type 1 receptors [44]. TRIB2 interacts with NF-κB and with substrates of the ubiquitin-proteasome system (TCF4, β-catenin, C/EBPα and CDC25B/C) [45].

Little is known about the involvement of UBTF in cancer progression. In melanoma, UBTF has been shown to act as cell cycle regulator. UBTF facilitates the transcription of G-protein-coupled receptor kinase-interacting protein 1, thereby activating MEK1/2-ERK1/2 signaling [46]. UBTF is also involved in the regulation of TOR signaling [47]. Our data confirm that in gliomas, during the formation of neurospheres, the activation of UBTF commonly leads to an increase in the mRNA level of the key component of mTORC2—RICTOR (Figure 4 and Figure 5). RICTOR, through cellular signaling downstream of receptor tyrosine kinase (PI3K/AKT/mTOR), is actively involved in cytoskeleton assembly, cancer invasion processes, proliferation, metastasis and poor prognosis [48].

Thus, the activation of UBTF-dependent genes potentially modulates Notch-, TCF4- and BMP-signaling, as well as the mTORC2 pathway.

#### 3.3.3. ZEB1 Transcription Factor mRNA- and ZEB1-Controlled Genes

The list of UBTF-upregulated transcripts also includes ZEB1 mRNA (Figure 5, Appendix A). TF ZEB1 plays an important role in GBM progression by acting as a pro-tumoral effector, and ZEB1 expression in GBM predicts the shorter survival and poor response to temozolomide [49,50].

Importantly, ZEB1-controlled transcripts are indexed in the Enrichr library “ENCODE and ChEA Consensus TFs from ChIP-X”. Despite the fact that the level of ZEB1 mRNA is statistically significantly and unidirectionally elevated in NS cultures (Figure 4), we did not reveal a significant enrichment in ZEB1-dependent transcripts, as shown in the top 10 TFs list (Table 3). Thus, an increase in the level of ZEB1 mRNA does not necessarily lead to the common and large-scale activation of ZEB1-dependent genes in glioma cells.

#### 3.3.4. Activation of NFE2L2-Dependent Genes

NFE2L2 (NFE2 Like BZIP Transcription Factor 2, Nrf2) is a TF that is mainly involved in iron metabolism, oxidative defense, and redox imbalance in ferroptosis [51]. The mRNA level of the NFE2L2, as well as the mRNA level of the SOX2 and UBTF, did not undergo unidirectional changes under the conditions of NS formation (Figure 4). In light of this, in Enrichr data, we observed the statistically significant enrichment of NFE2L2-controlled transcripts in lists of all particular MN/NS pairs, as well as in the list of common upregulated genes (Table 3 and Appendix A, Figure 5).

The list of genes controlled by NFE2L2 includes PLAUR, HIPK2, and TCF4 (Figure 5, Appendix A). PLAUR encodes the urokinase receptor (uPAR). PLAUR has been shown to promote glioblastoma NS cell survival and is associated with a more aggressive mesenchymal subtype of glioblastoma tissue [52]. HIPK2 is involved in TP53-mediated cellular apoptosis and the regulation of the cell cycle [53]. TCF4 in glioma cells mainly acts in the Wnt/β-catenin-signaling pathway and interacts with TRIB2 [45]. The transcription factor TCF4, forming the complex with β-catenin, binds with Akt2 promoter and activates Akt signaling cascades [54].

Recently, the NFE2L2 mRNA level has been shown to correlate with poor prognosis in patients with low-grade glioma [55]. NFE2L2 is involved in mediating TMZ glioblastoma resistance via MMP-2 [56]. NFE2L2 is also directly involved in the stabilization of the hybrid epithelial/mesenchymal state in RT4 urinary bladder transitional cell papilloma and UM-UC-1 bladder transitional cell carcinoma. Experimental–computational analysis revealed that the Nrf2(NFE2L2)-EMT-Notch1 network coordinates cancer cells in the migrating front during collective migration [57].

Thus, our results and the literature data indicate a significant role of the transcription factor NFE2L2, not only in the regulation of the formation of glioma neurospheres, but also in the stabilization of the hybrid epithelial–mesenchymal phenotype, which promotes the migration and invasion of cancer cells.

#### 3.3.5. Activation of STAT3 TF- and STAT3-Dependent Genes

STAT3 (signal transducer and transcription activator 3) plays a critical role in the pathogenesis of gliomas, immune suppression, immune cell tolerance, the proliferation and migration of glioma cells, promoting angiogenesis, and the stemness maintenance of CSCs. The activation of STAT3 is induced by cellular plasma membrane receptors, such as growth factors receptors (EGFR, PDGFR, FGFR, etc.) as well as cytokines receptors [58]. It is worth noting that the role of STAT3 is twofold—it regulates both oncogenes and tumor suppressor genes, so it can stimulate or inhibit oncogenesis depending on its interaction with various signals in the oncogenic environment and/or the presence of two splicing isoforms [59].

In spite of the fact that the “ENCODE and ChEA Consensus TFs from ChIP-X” Enrichr library contains special gene lists of STAT3 indicator genes—” STAT3 ENCODE” and “STAT3 CHEA”—we did not find STAT3 to be a statistically significant TF in the lists of the top 10 common (overlapping) or particular TFs (Table 3). At the same time, enrichment in STAT3-dependent genes is observed when analyzing a list of 203 unidirectionally activated genes in “ChEA 2016” or “ENCODE TF ChIP-seq 2015” libraries separately (Appendix A). Here, we used data for STAT3 controlled genes from the Enrichr library “ENCODE TF ChIP-seq 2015” concerning the overlap with the “STAT3 HeLa-S3 hg19” gene list (Appendix A, Figure 5).

The list of common upregulated STAT3-controlled genes includes KLF9, BCL6, MCL1, ITGA2 and ERRFI1 (Figure 5). The KLF9 is known to suppress Notch1 signaling and inhibit glioblastoma-initiating stem cells [60]. In turn, BCL6 is known to be a glioma-promoting gene and a biomarker whose activation correlates with the clinical grade. BCL6 protein regulates CSC self-renewal through Notch signaling [61]. Such data may indicate the presence of a negative feedback loop between the activation of Notch signaling by upregulated STAT3-controlled genes and the maintenance of stemness. PTGS2 activates the NF-κB signaling pathway, leading to tumor cell proliferation and tolerance to radiotherapy [62]. MCL1 modulates cell division through interactions with cell cycle regulators, acts as a molecular switch for double-strand break DNA repair, regulates autophagy and mitophagy, and modifies calcium homeostasis at the ER and mitochondrial membranes [63]. MCL1 silencing has been shown to lead to the senescence and apoptosis of glioma cells through the inhibition of the PI3K/Akt signaling pathway [64]. ITGA2 (encodes integrin α2 subunit, CD49b) plays a role in cancer cell migration, cancer stemness and differentiation [65]. Importantly, ERRFI1 mediates EGFR endocytosis and lysosomal degradation, and promotes the ubiquitination and degradation of the receptor, which qualifies it as a bona fide feedback inhibitor of the EGFR signal transduction pathway [66]. Thus, the STAT3-dependent (as well as SOX2- and TCF3-dependent) increase in ERRFI1 expression provides the common glioma path for the regulation of EGFR signaling.

#### 3.3.6. Transcription Factor TCF3 and TCF3-Dependent Genes

Our data show that TCF3 is represented in the list of the top 10 activated TFs of all analyzed glioma NS, except BR2 (Table 3). Accordingly, the enrichment of the glioma’s overlapping activated genes in those controlled by TCF3 is not highly statistically significant (Enrichr *p*-value < 0.01; adjusted *p*-value > 0.05; Appendix A). However, a growing body of new data on TCF3 points to the high importance of genes controlled by TCF3 and TCF3 itself in the development and invasion of glioma cells.

TCF3 is a member of the E protein family of the helix–loop–helix transcription factors belonging to the Tcf/Lef family of Wnt signaling effector molecules. TCF3 is involved in neuronal differentiation, and is considered as an intracellular inhibitor of pluripotent cell self-renewal that acts by limiting the sustained levels of self-renewal factors [67]. The overexpression of TCF3 has been detected in several types of human cancers, including Wilms’ tumor, breast cancer, renal carcinoma and embryonal carcinoma [68]. The TCF3-β-catenin complex activates the classic Wnt signaling pathway, regulates cell proliferation, and is closely associated with the onset and development of tumors. TCF3 also promotes glioma development through PI3K/Akt and MAPK-Erk signaling pathways [69].

Our data show that the relative level of TCF3 mRNA is downregulated in NS (Figure 4). With that from the set of upregulated genes 17 and 19 TCF3 controlled transcripts are indexed in the “TCF3 ENCODE” and “TCF3 CHEA” Enrichr libraries, respectively (Figure 5, Appendix A), including MCL1, TCF4 mentioned above and ATP1A1, ZMYM2, IRF2BL2, SAT1, WWTR1, SATB1 and others.

The subunit of Na^+^/K^+^-ATPase a1 (ATP1A1), which is overexpressed in GSCs, is considered as a new therapeutic target for gliomas [70]. ZMYM2 promotes the association of the BRCA1 factor with double strand breaks, thus playing an important role in DSB mainly through homologous recombination [71]. It was shown that IRF2BPL drives the ubiquitylation and degradation of β-catenin, which points to a reverse loop in the regulation of the Wnt signaling pathway [72]. The overexpression WWTR1, a transcriptional coactivator with the PDZ-binding motif, leads to tumor proliferation and CSC renewal [73]. Phosphorylated special AT-rich sequence-binding protein 1 (SATB1) is a TF associated with the progression and poor prognosis of glioma [74]. SATB1 knockdown is known to affect important oncogenes, including Myc, Bcl-2, Pim-1, EGFR, β-catenin, and survivin, with molecules involved in the cell cycle, EMT, and cell adhesion. It was found that there is a positive feedback loop between mediators of the Wnt signaling pathway TCFL2/β-catenin and SATB1: the maintenance of Wnt signaling by SATB1 and the induction of SATB1 expression through the activation of Wnt signaling. This indicates the functioning of reverse regulation between the transcription factors of the TCF family and the transcription factor SATB1. The feedback between TCFs and SATB1 can potentially be considered as an essential component of the transcriptional network that regulates the formation of neurospheres by glioma cells, as well as the maintenance of the stemness, migration, and invasion of gliomas.

In general, the activation of TF TCF3 and elevated levels of TCF3-dependent transcripts is associated with the regulation of Wnt-, PI3K/Akt and MAPK-Erk signaling pathways.

#### 3.3.7. Repression of MYC- and MAX-Controlled Genes

Members of the MYC family of TFs play a critical role in the regulation of a wide range of biological processes, including metabolic processes, translation, cell proliferation, stemming, and neoplastic transformation [75].

It can be seen that the level of MYC mRNA, as well as the level of MAX mRNA, changed in different directions during the incubation of glioma cells under conditions of NS formation (Figure 4). At the same time, the list of genes with a reduced expression in neurospheres is statistically significantly enriched in those controlled by MYC/MAX (Table 3 and Appendix A). The list of MYC/MAX-controlled downregulated genes includes: genes of ribosomal proteins—RPL18A, RPL19, RPS12, RPS15 and others; translation factors—EEF2 and EIF3K; mitochondrial proteins—MRPL41, DNAJC11, GLRX5, TIMM44, TIMM13 and VDAC2 (Figure 4 and Figure 7, Appendix A).

Thus, the formation of neurospheres by glioma cells is accompanied by a decrease in the level of mRNA sets encoding translational and metabolic factors, including mitochondrial ones.

Downregulated genes common to different glioma cell cultures are enriched in transcripts controlled by MYC and MAX. At the same time, the genes controlled by MYC and MAX overlap significantly with each other (Figure 7). This can be explained by the formation of the MYC–MAX heterodimer, which regulates the transcription of target genes. Signaling pathways and cellular processes associated with the downregulation of MYC/MAX-controlled gene expression in NS gliomas are summarized in Figure 8.

### 3.4. Cellular Processes and Signaling Pathways Underlying the Formation of Glioma Neurospheres

KRAS signaling, TNF-alpha signaling via NF-kB, TGF-beta signaling and EMT are the common processes determined using Enrichr in the “MSigDB Hallmark 2020” library for upregulated gene sets (Table 4). Confirming the data of the transcription factor analysis (Table 3), the results of cellular processes and signaling pathways indicate a decrease in the level of MYC-dependent genes—the downregulation of Myc-controlled transcripts (“Myc Targets V1” and “Myc Targets V2” in Table 4). Data on cellular processes and signaling pathways obtained using Enrichr are confirmed through an independent analysis of the gene lists on the GSEA platform (Appendix A).

#### 3.4.1. KRAS Signaling

KRAS is a member of the RAS protein family, which is known to be a key participant in EGF- and FGF-receptor signal transduction pathways [76]. In our data, the relative level of KRAS mRNA changed in different directions in glioma cell lines (Figure 4). We found four mRNAs of the RAS superfamily members, RAP1A, RAB8B, RAB31 and ARL6, which were unidirectionally and statistically significantly activated in all analyzed NS cell cultures. Three of the four activated mRNAs of the RAS superfamily encode proteins involved in intracellular vesicle transport—RAB8B, RAB31 and ARL6. Importantly, RAB31 is a key protein that is directly involved in EGFR delivery to late endosomes, which leads to receptor degradation [77]. Thus, an increase in the expression of RAB31 provides a feedback regulation of EGFR signaling.

In addition to members of the RAS superfamily, our data highlight the following upregulated genes whose products are involved in KRAS (Ras) signaling processes: BMP2, TRIB2, ITGA2 (see above) and ADAM17 (Appendix A, Figure 4). ADAM17 metallopeptidase participates in processing EGFR ligands, and plays a prominent role in the activation of the Notch signaling pathway [78].

#### 3.4.2. TGF-Beta Signaling and TNF-Alpha Signaling via NF-kB

Both the TGF-beta pathway and the TNF-alpha signaling pathway through NF-kB are important molecular processes for glioma growth and invasion [79,80].

Our Enrichr-processed RNA-seq data delineate a list of NS-activated genes whose products are involved in TNF/NF-kB/TGFb regulatory pathways, including BMP2, PLAUR, RAB31, BCL6 and MCL1 (see above). The list also includes IER3 (Appendix A, Figure 4). The IER3 encodes a protein that not only participates in the innate immune response (and TNF/NF-kB), but also modulates the MAPK/ERK and PI3K/Akt pathways [81].

#### 3.4.3. Epithelial to Mesenchymal Transition

EMT is considered to be the main cellular process providing the invasion, metastasis and resistance of cancer cells to therapy [22,80].

We determined that the formation of NS by glioma cells includes the significant and unidirectional activation of genes from the EMT group of the Enrichr library “MSigDB Hallmark 2020” (Table 4 and Appendix A). The list of common overlapping transcripts of NS related to EMP includes integrins ITGA2, ITGAV and PLAUR (see above). Other indicators of EMT activation in glioma NS include MMP1, TIMP1 and PLOD2 (Figure 4, Appendix A). *MMP1* encodes the zinc-dependent endopeptidase MMP-1, a key protease in ECM degradation and cell invasion. Previously, MMP-1 and plasminogen urokinase activator (uPA) were identified as potential STAT6 targets responsible for GBM cell invasion [82]. PLOD2, which encodes membrane-bound procollagen lysyl hydroxylase, has been shown to enhance proliferation, invasion, attachment-independent growth and promote the progression of GBM [83].

#### 3.4.4. GO Annotations and KEGG Pathways

Enrichr analysis of up- and downregulated transcripts in the library “GO Biological Process 2021” confirms conclusions about the key processes that determine the formation of glioma neurospheres. Thus, three top ranked GO biological process terms are: “regulation of differentiation of mesenchymal stem cells”, “positive regulation of stem cell differentiation”, and “positive regulation of cell differentiation” (Appendix A). In light of this, downregulated genes are enriched in GO terms: “SRP-dependent cotranslational protein targeting to membrane”, “translation”, “cytoplasmic translation”, and “cotranslational protein targeting to membrane” (Appendix A).

From the list of upregulated genes indexed in GO terms “mesenchymal stem cell differentiation” and “positive regulation of stem cell differentiation”, we should outline the following genes: PDGFRA, SOX9 and SOX5 (Figure 4, Appendix A). PDGFRA encodes receptor tyrosine kinase, which plays a role in glioblastoma initiation and progression and is also known as an indicator of GBM pro-neural subtype [84]. SOX5 and SOX9, as members of the SOX family, are implicated in the development and maintenance of CNS tumors [85].

ADAM17 and MAML3 are indexed in the KEGG “Notch signaling pathway”, and BMP2 and SMAD5 are indexed in the KEGG “TGF-beta signaling pathway” (Appendix A), which confirms our data on the involvement of these processes in the formation of glioma neurospheres.

Downregulated genes in glioma neurospheres, RPL and RPS families (RPLP1, RPSA and others), are indexed in the GO term “translation” (GO:0006412), as well as in the “KEGG 2021 Human Ribosome” libraries (Appendix A). This highlights the involvement of the MYC/MAX family of transcription factors in the regulation of NS formation.

The activation and, separately, the suppression of gene expression through transcription factors and the impact on biological processes and signaling pathways in glioma neurospheres are summarized in Figure 6 and Figure 8.

### 3.5. Validation of NGS Data with qRT-PCR

In order to confirm RNA sequencing data using an independent approach, we performed qRT-PCR analysis of eight randomly selected transcripts, CXCL1, ERRFI1, NFKBIA, NRP2, PDGFRA, SOX2, TRIB2, and ZEB1. Relative transcript levels were analyzed via real-time PCR using RNAs from independent samples of adherent and neurosphere glioma cultures. For all analyzed genes, it was determined that the linear regression between the relative expression obtained from the DESeq normalized gene values and the qPCR genes levels were characterized by Pearson R^2^ from 0.60 to 0.97 (Appendix A). Thus, the high correlation of NGS data with independent qRT-PCR results generally confirms our findings.

## 4. Discussion

The conditions for the neurospheres’ formation are the standard cultivation for both enriching neural stem cells and brain tumor stem cells, as well as for investigating the molecular features of glioma. One of the most commonly used components for neurosphere formation are EGF and bFGF; they allow the generation of spheres that are much more similar to human glioblastoma than to tumors initiated by glioma cell lines [86]. Previously, there have already been attempts to investigate the molecular mechanisms that occur in cells under the influence of serum-free media and the above components, and, accordingly, during the transition from adherent forms to neurospheres. Several groups announced that GBM spheres created in serum-free medium with EGF and bFGF had the ability to be highly invasive, to proliferate and retain their multipotency; that is, they can differentiate into cells expressing astrocytic, oligodendroglial and neuronal markers, through binding to the EGF and FGF receptors and triggering the corresponding signal transduction pathways [6,7,10].

A number of studies have conducted the transcriptome analysis of glioblastoma stem cells, explants and neurospheres [87,88,89,90], which demonstrates the relevance of such models for preclinical investigations. At the same time, only a few studies have compared the metabolomic, transcriptomic, and proteomic data of glioma cells cultivated under adherent culture and neurosphere conditions. In 2014, attempts were made to identify common and distinct proteins of adherent cells and neurospheres using proteomic analysis with nano-LC/Q-TOF MS [91]. It was shown that the proteomic patterns of adherent cells and neurospheres are different; the former are characterized by patterns of cell adhesion and invasion, whereas the latter are characterized by proteins involved in the cell cycle and protein metabolism. Baskaran S. et al. compared genome copy number variations with changes in gene expression in three patient-derived GBM cell cultures propagated in conditions that are quite similar to NS formation. Consistent transcriptional changes between earlier (<10) and later (<30) passages of GBM cultures showed the induction of pathways associated with GSEA ribosomal biogenesis, oxidative phosphorylation, tricarboxylic acid cycle, mTOR-signaling, and hypoxia [5]. Peixoto J. et al. analyzed metabolic discrepancies between the GBM cell line (U87) and a patient-derived GBM stem-like cell line (NCH644) exposed to neurospheres or monolayer culture conditions using transcriptomics and metabolomics. It was found that arginine biosynthesis was the most significantly regulated pathway in neurospheres and that GSCs may exhibit classical auxotrophy [92]. Thus, a comprehensive study of the transcriptome during the transition from adherent cell cultures to neurospheres under the influence of growth factors (EGF and bFGF) is currently lacking.

This is why aim of this work was to perform an exhaustive study of differences in the transcriptome data of immortalized and patient-derived cell cultures during the transition from adherent cultures to neurospheres using the most commonly used bioinformatics tools. In the present work, we used five cell cultures, three of which are patient-derived gliomas—BR1, BR2, and MG1—and two of which are immortalized GBM cell lines—U87 MG and U343 MG. We cultivated the cells in conditions of NS formation using four to eight passages and compared the transcriptomes of MN and NS cells in each particular MN/MS pair, as well as in terms of the overlapping set of upregulated and downregulated genes.

In order to describe TFs involved in the adaptation of the transcriptome of glioma cells to the conditions of culturing neurospheres, we used Enrichr gene set enrichment analysis [31]. We observed that NS formation is accompanied by the activation of transcription factors SOX2, UBTF, TCF3 and NFE2L2 (Table 3). We also detected that STAT3-controlled genes were upregulated in glioma NS (Appendix A). The sets of transcripts controlled by MYC and MAX (Figure 7) were suppressed in NS (Table 3). The genes controlled by SOX2, UBTF, NFE2L2, TCF3 and STAT3 (Figure 5) are involved in the processes of EMT, the regulation of mesenchymal stem cell differentiation, the invasion and migration of GBM and others (Figure 6), while MYC/MAX-dependent downregulated genes (Figure 7) are involved in translation, focal adhesion apical junction and other processes summarized in Figure 8.

Interestingly, we found three commonalities for all analyzed glioma feedback regulators of the EGFR and FGFR signaling pathways: SPRY4, ERRFI1 and RAB31 (Figure 9).

ERRFI1 has been found to be significant tumor suppressor gene and is frequently deleted, mutated or downregulated in various types of cancer, including glioblastomas [66]. ERRFI1 overexpression has been shown to reduce proliferation in GBM cells by binding EGFR to Syntaxin-8 and targeting internalized EGFR to late endosomes for degradation, while knockdown of ERRFI1 expression resulted in increased tumor invasion [68,79,93]. RAB31 genes products involved in EGFR endocytosis and lysosomal degradation also as ERRFI1. Moreover, RAB31 is on the list of genes with the greatest influence on the development of the highest-grade astrocytoma, glioblastoma multiforme. The genes on this list can predict tumor status with 96–100% confidence using logistic regression, cross-validation, and support vector machine analysis [94]. Earlier was shown the tumor-suppressing role in GBM-derived cell lines of the Spry4 protein which has important functions in many receptor tyrosine kinase-mediated signal transduction cascades [93]. It specifically interferes with MAPK-ERK activation and phospholipase C-induced pathway, affects the PI3K pathway [41]. Thus, the activation of the expression of these genes can be considered as naturally functioning processes of control of the epithelial–mesenchymal transition and cancer stemness, which is common for glioma cells. In general, the activation of SPRY4, ERRFI1, and RAB31 can be used for developing new approaches to glioma therapy.

## 5. Conclusions

In this work, we focused on key transcription factors and the genes controlled by them during the formation of neurospheres in glioma cell cultures. In the transition from the adherent cell model to neurospheres, when using serum-free media and different growth factors as components, it is necessary to clearly understand the changes that occur at the cellular level. This is of fundamental importance, as it helps us to understand the molecular mechanisms of the activation of signal transduction pathways that trigger oncogenesis, as well as practical importance in the development of both the targeted drugs for the treatment of glioma and the diagnostic signatures that have a prognostic effect.

In general, our data emphasize the role of transcription factors, the participation of which has already been described earlier in the processes of epithelial–mesenchymal and/or neuro-mesenchymal transition—SOX2, STAT3, and TCF3. In addition, we revealed the involvement of transcription factors UBTF and NFE2L2, which, to the best of our knowledge, have not been previously associated with the formation of aggressive forms, invasion, and metastasis of gliomas. The identification of the SPRY4, ERRFI1, and RAB31 as genes whose natural activation provides the inverse regulation of the processes of neurosphere formation can be used for creating new strategies of suppressing the invasion and progression of gliomas.

## Figures and Tables

**Figure 1 cells-11-03106-f001:**
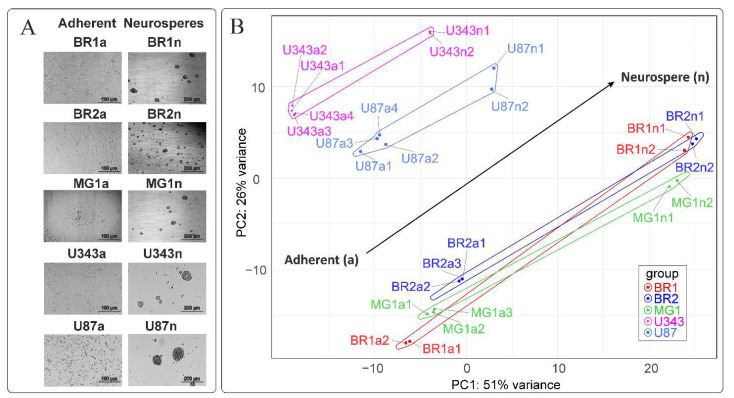
(**A**) Representative images of MN (adherent) and NS glioma cells before cell collection for NGS RNAseq analysis. (**B**) Principal component analysis of DESeq2 normalized, variance stabilizing transformed (VST) gene expression data. Cell line-specific PC1:PC2 points are annotated with cell line-defined envelopes. The black arrow shows the general trend of PC1:PC2 transition from the MN state to the NS state.

**Figure 2 cells-11-03106-f002:**
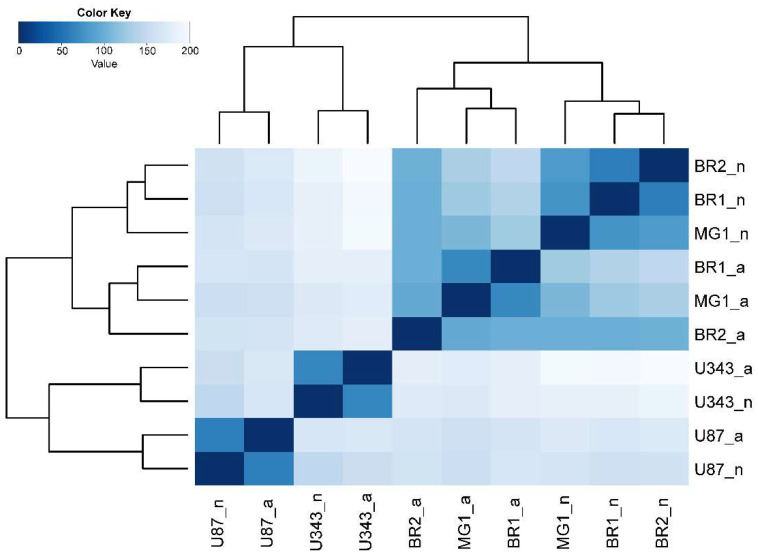
Heatmap and sample tree of Euclidean distances of RNAseq variance stabilizing transformed (VST) gene expression data of glioma cell cultures. The complete agglomeration method for clustering was used.

**Figure 3 cells-11-03106-f003:**
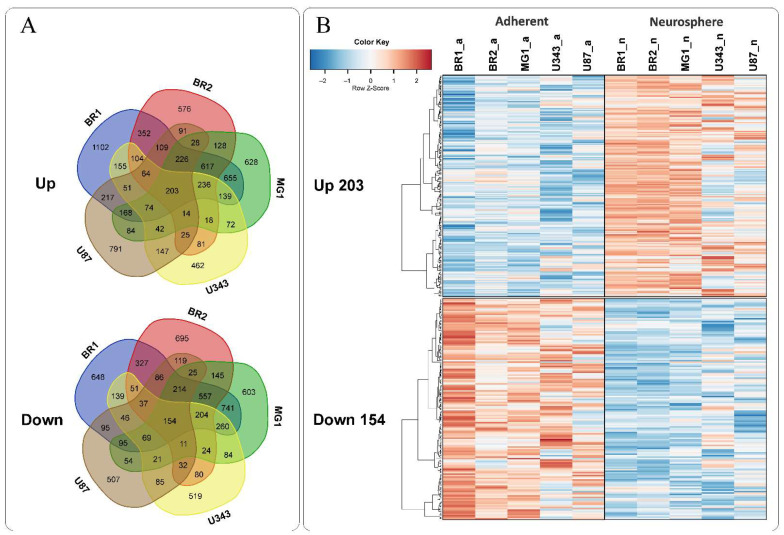
(**A**) Venn diagrams showing intersections of gene sets in selected glioma cultures, separately for genes with increased (Up) and decreased (Down) levels compared to the corresponding MN cells. (**B**) Heatmaps of 203 commonly upregulated and 154 commonly downregulated genes in NS vs. MN glioma cells.

**Figure 4 cells-11-03106-f004:**
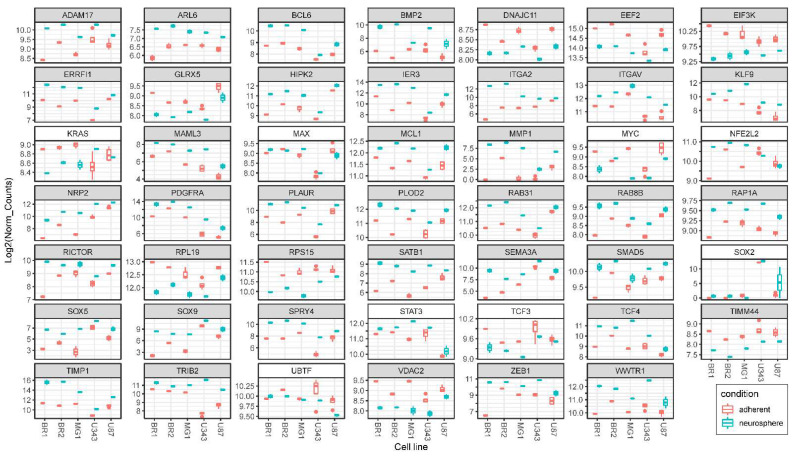
Differences in relative mRNA levels in adherent glioma cultures and neurospheres. Box plots of DESeq2 normalized expression values of mRNAs grouped by glioma cell lines and colored red for monolayer (adherent) cultures and cyan for corresponding NS. Plots with gray shaded headers represent data for differentially expressed genes that meet the criteria DESeq2 padj < 0.05, Log2FoldChange > 0 for upregulated or log2FoldChange < 0 for downregulated transcripts (for all MN/NS pairs in one direction—synchronously “up” or “down”). Plots with white shaded headers represent mRNA levels of TFs and KRAS that do not meet the criteria for differentially expressed genes (discussed in text).

**Figure 5 cells-11-03106-f005:**
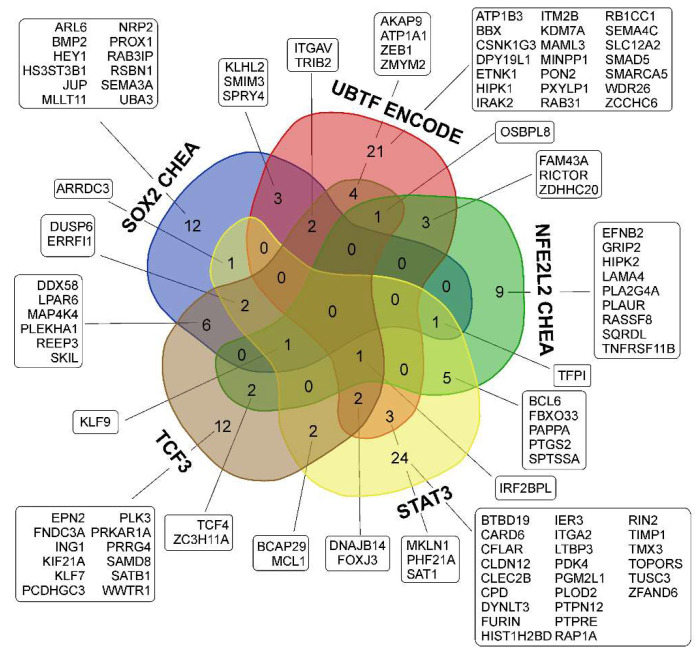
Venn diagram showing intersections of sets of glioma NS upregulated genes controlled by SOX2, UBTF, NFE2L2, STAT3 and TCF3 transcription factors. Glioma NS upregulated genes analyzed using Enrichr library “ENCODE and ChEA Consensus TFs from ChIP-X”. STAT3 controlled genes from the Enrichr library “ENCODE TF ChIP-seq 2015” concerning overlap with “STAT3 HeLa-S3 hg19” gene list. TCF3-dependent genes from the union of “TCF3 ENCODE” and “TCF3 CHEA” Enrichr libraries “ENCODE and ChEA Consensus TFs from ChIP-X”.

**Figure 6 cells-11-03106-f006:**
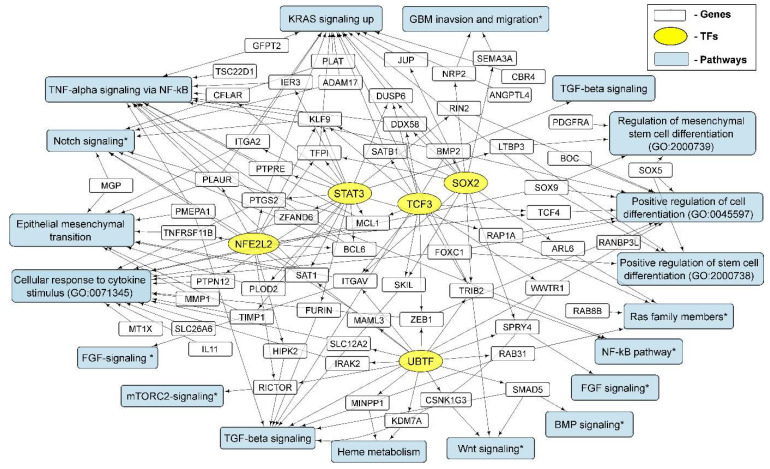
The relationship between the activation of transcription factors SOX2, UBTF, NFE2L2, TCF3 and STAT3; individual activated genes and signaling pathways; and biological processes characteristic of the formation of NS glioma cells. The results of the analysis of gene sets using the Enrichr “GO Biologic Process 2021” and “KEGG_2021_Human” libraries are summarized with the addition of recently published data presented in the text (marked with *).

**Figure 7 cells-11-03106-f007:**
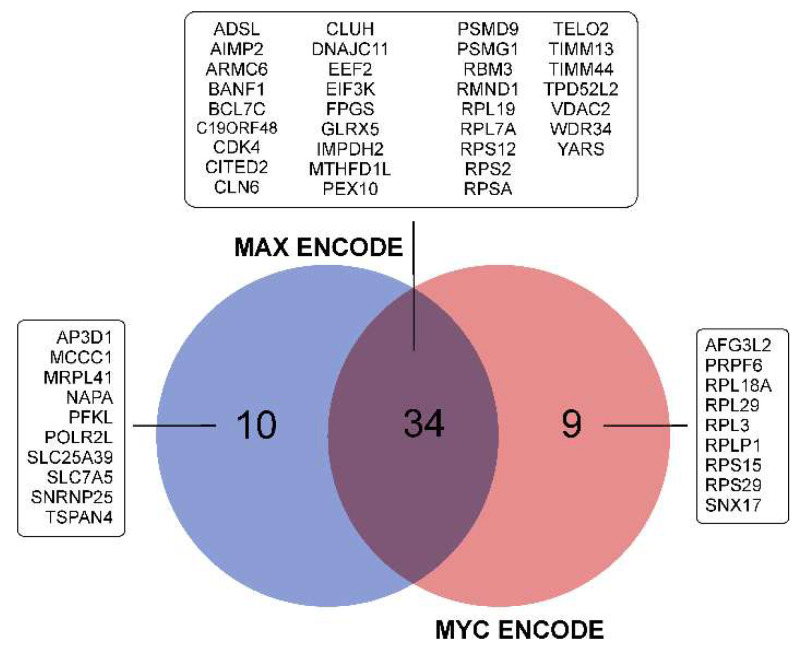
Venn diagram showing intersections of sets of glioma NS-downregulated genes controlled by MYC and MAX transcription factors. Illustrated Enrichr results of glioma NS downregulated genes analyzed using the “ENCODE and ChEA Consensus TFs from ChIP-X” library.

**Figure 8 cells-11-03106-f008:**
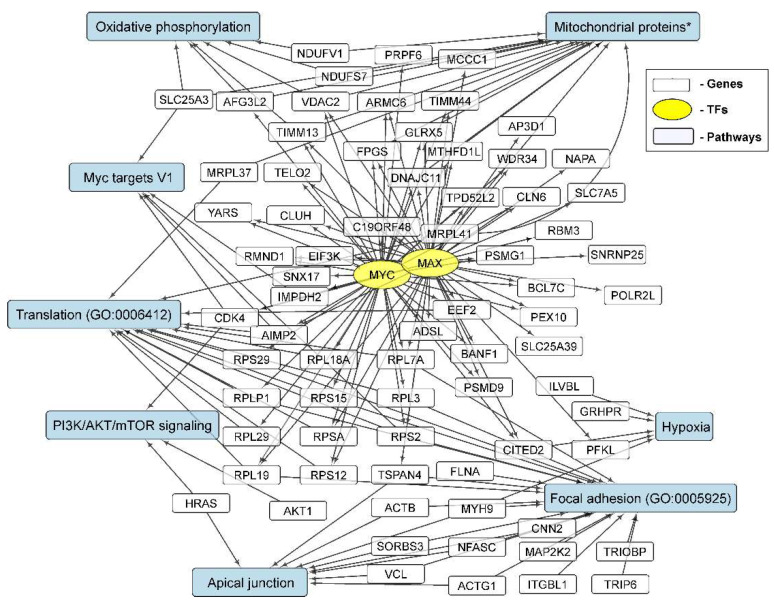
Relationships between MYC and MAX transcription factor repression, particular downregulated genes and signaling pathways, and biological processes common for glioma NS cell cultures formation. Summarized results of gene sets analysis with the Enrichr libraries “GO Biological Process 2021” and “KEGG_2021_Human”, as well as recently published data presented in the text (marked with *).

**Figure 9 cells-11-03106-f009:**
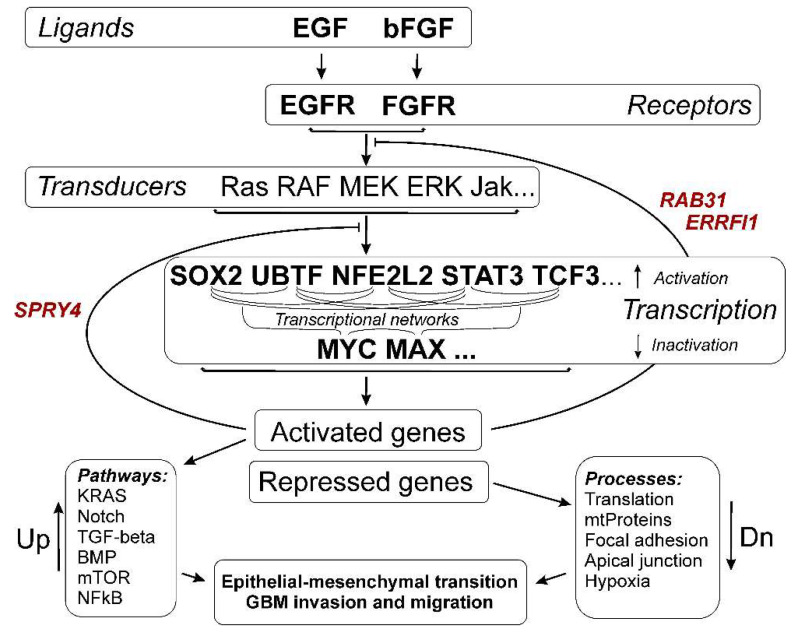
General scheme of processes in glioma cells under conditions of neurosphere formation. The *SPRY4, ERRFI1*, and *RAB31* genes are indicated, the activation of which provides inverse regulation of both the formation of neurospheres and the processes of tumor progression associated with them, including EMT, invasion, and migration of cancer cells.

**Table 1 cells-11-03106-t001:** Characteristics of human brain tumor cell cultures and RNA sequencing data.

Cell Culture	Histological Characteristic	Culture Conditions	NGS-Library *	Number of Replicates	Number of NGS-Sequencing Reads (10^6^) ***
**BR1**	GBM	MN	BR1a	2	22.94
		NS	BR1n	2	19.45
**BR2**	Diffuse astrocytoma	MN	BR2a	3	35.47
		NS	BR2n	2	19.68
**MG1**	GBM	MN	MG1a	3	34.10
		NS	MG1n	2	18.98
**U343 ****	GBM	MN	U343a	4	46.60
		NS	U343n	2	24.10
**U87 ****	GBM	MN	U87a	4	46.35
		NS	U87n	2	22.95

* The names of the NGS libraries used in this article correspond to the names of the cell cultures. ** Also known as U-343-MG and U-87 MG human glioblastoma cell lines. MN—monolayer (adherent); NS—neurospheres. *** Total number of sequencing reads for all replicates of the NGS library.

**Table 2 cells-11-03106-t002:** Differentially expressed genes in MN/NS pairs of glioma cell cultures.

**Particular MN/NS Pairs**					
	BR1	BR2	MG1	U343	U87
**Up ***	4472	2872	3332	1887	2334
**Down ***	3723	2761	3261	1816	1650
**Common (Overlapping) Genes**					
**All Up**			203		
Up/Down **			121		
**All Down**			154		

* Genes selected by DESeq2 padj < 0.05: Log2FoldChange > 0 for upregulated or log2FoldChange < 0 for downregulated. ** Transcripts with padj < 0.05 and with non-unidirectional expression changes in different MN/NS pairs.

**Table 3 cells-11-03106-t003:** Transcription factors controlling gene expression during NS formation determined using Enrichr. Top 10 Enrichr records (library “ENCODE and ChEA Consensus TFs from ChIP-X”), ordered by descending *p*-value (with adjusted *p*-value < 0.05), for each particular MN/NS pair as well as for the list of overlapping up/downregulated genes. Common top transcription factors for comparison pairs are highlighted in color.

Rank	Overlapped	BR1	BR2	MG1	U343	U87
	**Upregulated**					
1	SOX2 CHEA	UBTF ENCODE	UBTF ENCODE	UBTF ENCODE	SOX2 CHEA	UBTF ENCODE
2	UBTF ENCODE	NFE2L2 CHEA	NFE2L2 CHEA	NFE2L2 CHEA	SUZ12 CHEA	AR CHEA
3	FOXA2 ENCODE	AR CHEA	TAF1 ENCODE	SOX2 CHEA	NFE2L2 CHEA	NFE2L2 CHEA
4	NFE2L2 CHEA	SOX2 CHEA	CREB1 CHEA	SUZ12 CHEA	AR CHEA	SOX2 CHEA
5	TP53 CHEA	SMAD4 CHEA	SOX2 CHEA	ZBTB7A ENCODE	TCF3 CHEA	ZNF384 ENCODE
6	SALL4 CHEA	TCF7L2 ENCODE	BRCA1 ENCODE	GATA1 CHEA	TP63 CHEA	GATA1 CHEA
7	AR CHEA	FOXA2 ENCODE	PPARG CHEA	AR CHEA	UBTF ENCODE	SMAD4 CHEA
8	TCF3 ENCODE *	CHD1 ENCODE	CREB1 ENCODE	FOXA2 ENCODE	GATA2 CHEA	TCF3 ENCODE
9	TCF3 CHEA *	ZNF384 ENCODE	RUNX1 CHEA	TCF3 ENCODE	NANOG CHEA	CTCF ENCODE
10	VDR CHEA *	TCF3 ENCODE	GATA2 CHEA	ESR1 CHEA	STAT3 CHEA	TCF3 CHEA
	**Downregulated**					
1	MYC ENCODE	NFYB ENCODE	USF1 ENCODE	E2F4 ENCODE	USF1 ENCODE	E2F4 ENCODE
2	MAX ENCODE	MAX ENCODE	USF2 ENCODE	MAX ENCODE	MAX ENCODE	E2F6 ENCODE
3	MYC CHEA	MYC ENCODE	KLF4 CHEA	MYC ENCODE	USF2 ENCODE	MAX ENCODE
4	USF2 ENCODE	NFYA ENCODE	MAX ENCODE	TAF1 ENCODE	MYC ENCODE	MYC ENCODE
5	TAF1 ENCODE	TAF1 ENCODE	BHLHE40 ENCODE	NFYB ENCODE	GABPA ENCODE	BRCA1 ENCODE
6	USF1 ENCODE	BRCA1 ENCODE	E2F6 ENCODE	MYC CHEA	ZBTB7A ENCODE	TAF1 ENCODE
7	ATF2 ENCODE	GABPA ENCODE	CTCF ENCODE	E2F6 ENCODE	YY1 ENCODE	NFYB ENCODE
8	PML ENCODE	MYC CHEA	ZBTB7A ENCODE	NFYA ENCODE	ELF1 ENCODE	ATF2 ENCODE
9	NFYA ENCODE	E2F4 ENCODE	SMC3 ENCODE	SIN3A ENCODE	TAF1 ENCODE	CREB1 ENCODE
10	ZBTB7A ENCODE	YY1 ENCODE	NFYB ENCODE	E2F1 CHEA	CREB1 CHEA	YY1 ENCODE

* Adjusted *p*-value > 0.05.

**Table 4 cells-11-03106-t004:** Cellular processes and signaling pathways. Essential cellular processes and signaling pathways of NS formation determined using Enrichr. Top 10 Enrichr records (library “MSigDB Hallmark 2020”), ordered by descending *p*- value (with adjusted *p*-value < 0.05), for each particular MN/NS pair, as well as for the list of overlapping up/downregulated genes. Common processes and signaling pathways are highlighted in color.

# *	Overlapped	BR1	BR2	MG1	U343	U87
	**Upregulated**					
1	KRAS Signaling Up	Interferon Gamma Response	TNF-alpha Signaling via NF-kB	TNF-alpha Signaling via NF-kB	TNF-alpha Signaling via NF-kB	EMT **
2	TNF-alpha Signaling via NF-kB	Interferon Alpha Response	Interferon Gamma Response	Interferon Gamma Response	Cholesterol Homeostasis	UV Response Dn
3	TGF-beta Signaling	KRAS Signaling Up	EMT **	Interferon Alpha Response	Apoptosis	Cholesterol Homeostasis
4	EMT **	TNF-alpha Signaling via NF-kB	Inflammatory Response	KRAS Signaling Up	EMT **	KRAS Signaling Up
5	Apoptosis	EMT **	KRAS Signaling Up	Inflammatory Response	Hypoxia	TGF-beta Signaling
6	Coagulation	TGF-beta Signaling	Interferon Alpha Response	EMT **	p53 Pathway	TNF-alpha Signaling via NF-kB
7	Complement	Inflammatory Response	TGF-beta Signaling	TGF-beta Signaling	Inflammatory Response	Complement
8	Androgen Response	IL-6/JAK/STAT3 Signaling	IL-6/JAK/STAT3 Signaling	IL-6/JAK/STAT3 Signaling	KRAS Signaling Up	Hypoxia
9	Estrogen Response Early	IL-2/STAT5 Signaling	Apoptosis	Estrogen Response Early	Angiogenesis	Coagulation
10	Angiogenesis	UV Response Dn	IL-2/STAT5 Signaling	Angiogenesis	Estrogen Response Early	Androgen Response
	**Downregulated**					
1	Apical Junction	Myc Targets V1	Cholesterol Homeostasis	E2F Targets	Myc Targets V2	E2F Targets
2	Oxidative Phosphorylation	Oxidative Phosphorylation	Fatty Acid Metabolism	Myc Targets V1	Apical Junction	Oxidative Phosphorylation
3	Hypoxia	E2F Targets	p53 Pathway	G2-M Checkpoint	Mitotic Spindle	Myc Targets V1
4	Myc Targets V1	G2-M Checkpoint	Myogenesis	Oxidative Phosphorylation	EMT **	G2-M Checkpoint
5	Androgen Response	mTORC1 Signaling	mTORC1 Signaling	mTORC1 Signaling	Unfolded Protein Response	DNA Repair
6	PI3K/AKT/mTOR Signaling	DNA Repair	Adipogenesis	Mitotic Spindle	UV Response Dn	Mitotic Spindle
7	***	Adipogenesis	Mitotic Spindle	Glycolysis	Adipogenesis	Estrogen Response Late
8	***	Fatty Acid Metabolism	UV Response Dn	Myc Targets V2	PI3K/AKT/mTOR Signaling	p53 Pathway
9	***	Myc Targets V2	Hypoxia	Hypoxia	Estrogen Response Early	Adipogenesis
10	***	Cholesterol Homeostasis	Oxidative Phosphorylation	DNA Repair	Myc Targets V1	Myc Targets V2

* Rank of Enrichr records, ordered by descending *p*-value. ** EMT—Epithelial Mesenchymal Transition. *** Adjusted *p*-value > 0.05.

## Data Availability

RNAseq data of five glioma cell cultures in MN and NS state have been deposited at the SRA database under the accession code PRJNA869596. All the other data supporting the findings of this study are available within the article and Appendix A.

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
