# Peer review of "Transcriptome Changes in Glioma Cells Cultivated under Conditions of Neurosphere Formation"

_cells, 2022, doi:10.3390/cells11193106_

Round 1
Reviewer 1 Report
In this study, Vasileva et al perform transcriptional profiling of immortalized glioblastoma cell lines and patient derived cell lines under adherent and neurosphere-forming culture conditions. By comparing the transcriptomes of these cells, they identify a collection of transcription factors and related targets which are upregulated in neurospheres and involved in glioma cell stemness and invasive capacity. While the datasets produced here can be a useful resource for the community, there are several critical points that must be addressed with regards to potential confounding factors, statistical methods applied, missing data and over-interpretation of the results presented here.
Specifically:
- Key information about the glioma samples used to derive cell lines (BR1, BR2, MG1) is missing and should be provided, including molecular subtype (classical, proneural, neural, mesenchymal) and IDH mutational status. Are these samples treatment-naive? If not, what treatment did the patients receive prior to surgery? It is critical to have this information to control for potentially confounding factors in subsequent analyses.
- Were all patient-derived cell lines collected at the same passage? This should be indicated in a per-sample basis, since it can be a potential contributor to the transcriptional differences reported.
- Table 1. Does the number of sequencing reads indicate mean of all replicates, sum, other? This should be specified. If authors consider relevant to include this information, other quality-related information such as % of duplicated sequences, per sequence quality scores, etc should be included as well, since sequencing depth alone is not enough to prove technical reliability of an RNA-Seq experiment.
- In line 220, authors claim: "(...) the rate of their growth did not depend on the grade of the primary brain tumor", however they fail to provide experimental data to support this statement.
- Figure 1B: please correct labeling of BR1, BR2, MG1 samples in PCA plot; I assume from the text that all samples in the lower left part of the plot are "adherent" and should be labeled as "a", but they are all labeled as "n".
- In line 252, authors claim: "(...) NS formation has the common transcriptional features - gene patterns for all analyzed gliomas" but U87n and U343 cluster with their adherent counterparts and not with BR1n, BR2n, MG1n, which argues against this statement. Did authors refer exclusively to patient derived cell lines? Please, clarify.
- Many of the conclusions reached by the authors rely on the pathway enrichment analyses presented in tables 3 and 4. However, it is not clear whether authors are exclusively employing adjusted p-values as a threshold to consider these pathways significant. Since a high number of pathways are being tested simultaneously, p-values must be adjusted, and adjusted p-values must be used as thresholds of significance. It is also important to clarify whether gene sets composing each transcription factor-associated pathway (for example "SOX2 CHEA") exclusively contain targets that are upregulated by the specific transcription factor or if they also contain genes that are downregulated. This is relevant to interpret what enrichment of upregulated genes in NS vs MN in these pathways means.
- Expression (RNA-Seq counts) of selected genes for all the cell lines analyzed in the different culture conditions are reported in figure 4 and used to substantiate many claims along the paper, such as "(...) for 4 out of 5 analyzed cultures (except for MG1), an increase in SOX2 mRNA level in NS cell cultures was detected" (lines 302, 302). Is SOX2 significantly upregulated when running differential expression analyses by DESeq2 and comparing NS vs MN in all these cell lines? Same question applies for all the genes represented in this figure, please indicate whether they are significantly changed in the different cell lines and culture conditions.
- In lines 308-309, authors state: "It can be assumed that the observed activation of the SOX2 TF is determined to a greater extent by post-translational modifications of the SOX2 protein and/or its interaction with other factors, rather than by the level of its mRNA in glioma cells". The data from which the authors derive this conclusion could also be explained by different expression dynamics of SOX2 in MG1 vs the other cell lines or collection of the samples at a different passage, for example, and no additional experimental data is provided to substantiate this claim. Similar statements are presented along the results section of the paper but are not supported by experimental data, please rephrase to indicate that this is speculation or provide experimental proof.
- Lines 502-504: MYC and MAX dimerize to exert their function so it is expectable that that they have shared targets and this does not indicate that they "cooperate in the process of NS formation". Data in figure 4 does not show consistent changes in these two transcription factors in the different cell lines and conditions either.
- qPCR validation data should be included in the manuscript, represented comparing MN and NS from all the cell lines tested in this study and the potential differences should be statistically tested to validate the changes observed by RNA-Seq. It is also surprising that authors did not validate MYC/MAX, TCF3, STAT3, NFE2L2 or UBTF expression, since these are the TFs the manuscript mostly focuses on.
- In the discussion, the translational potential of SPRY4, ERRFI1 or RAB31 is discussed, have functional assays downregulating any of these genes in vitro been performed to prove that targeting any of them could be a promising strategy to prevent glioma progression?
- There are several very relevant studies that were overlooked by the authors (PMID: 19497285, PMID: 34675201 and PMID: 35303420); results presented here should be also discussed in the context of these published findings.
Minor comments:
- Abstract: please, define NGS.
- Abstract: please, explain what "the Enrichr gene sets" means or simply mention the kind of analysis performed (pathway enrichment).
- Figure 1A: please increase magnification or image resolution, cells are difficult to distinguish.
- Figure 2: please indicate in figure legend or methods which clustering method was used.
- Figure 3B: please indicate what green and purple mean in the heatmap.
- Table S3. Sheet name in the Excel file is in Russian, authors may want to translate it to English.
Author Response
Dear Reviewers and Editors,
Thank you for giving us the opportunity to submit a revised draft of my manuscript titled Transcriptome Changes in Glioma Cells Cultivated under Conditions of Neurosphere Formation to Cells. We appreciate the time and effort that you and the reviewers have dedicated to providing your valuable feedback on my manuscript. We are grateful to the reviewers for their insightful comments on my paper. We have been able to incorporate changes to reflect most of the suggestions provided by the reviewers.
Here is a point-by-point response to the reviewers’ comments and concerns. Please find below our response to reviewer’s comments.
Comments from Reviewer 1
- Key information about the glioma samples used to derive cell lines (BR1, BR2, MG1) is missing and should be provided, including molecular subtype (classical, proneural, neural, mesenchymal) and IDH mutational status. Are these samples treatment-naive? If not, what treatment did the patients receive prior to surgery? It is critical to have this information to control for potentially confounding factors in subsequent analyses.
Response: Thank you for pointing this out. Corrected. We added the information about the treatment and IDH-status into the subsection 2.2. Patient-derived cell cultures the sentence - “All samples were collected from treatment-naïve patients. According to the transcriptome data, the analyzed glioma cultures do not have mutations in the coding regions of the IDH1 and IDH2 mRNAs”.
Regarding the molecular subtype, it should be noted that the classification applies only to glioblastoma (GMB), but not to all gliomas [PMID 20129251]. Typically, the molecular subclasses of GBM – classical (CL), proneural (PN), neural (NL), mesenchymal (MES) – refer to GBM cell cultures by several criteria, including: genome abnormalities, DNA methylation status, and transcriptome profile. Moreover, Verhaak R et al [PMID 20129251], noted that "In contrast, attempts to detect comparable transcriptional subtypes in immortalized cell lines were uninformative (data not shown)." So the extrapolation of the classification to immortalized U87 and U343 cell lines may create an erroneous representation in the context of our data comparing cell cultures in a monolayer state and in the form of neurospheres.
Thus, in our work, we can only assume that the transcriptome profile of only two cultures, BR1 and MG1, corresponds to one of the CL, PN, NL, or MS types.
In order not to provide redundant data concerning only two cell cultures out of five, which do not contribute significant information about the remaining cell cultures, we decided not to consider the issue of classifying BR1 and MG1 in this article.
- Were all patient-derived cell lines collected at the same passage? This should be indicated in a per-sample basis, since it can be a potential contributor to the transcriptional differences reported.
Response: We added the sentences into subsection "2.3. Cell Culture for Neurosphere Formation": “U87 MG and U343 MG were collected at 3 passages. BR1 and BR3 were collected at 4 passages, MG1 – at 8 passages”.
- Table 1. Does the number of sequencing reads indicate mean of all replicates, sum, other?
This should be specified. If authors consider relevant to include this information, other quality-related information such as % of duplicated sequences, per sequence quality scores, etc should be included as well, since sequencing depth alone is not enough to prove technical reliability of an RNA-Seq experiment.
Response: Thank you for revealing the insufficiency of the table data description. We added to the notes of the Table 1: "*** Total number of sequencing reads for all replicates of the NGS-library."
With regard to the description of the sequencing data, here we report that the data have been deposited in the SRA archive with BioProject accession PRJNA869596. Currently we keep the data in the status "To be released". The data will be released as soon as we obtained "submitted" state of the article.
According to that we changed the "Data Availability Statement: RNAseq data of 5 glioma cell cultures in MN and NS state have been deposited at the SRA database under the accession code PRJNA869596. All the other data supporting the findings of this study are available within the article and its supplementary information files."
- In line 220, authors claim: "(...) the rate of their growth did not depend on the grade of the primary brain tumor", however they fail to provide experimental data to support this statement.
Response: We agree with reviewer. The sample is small, and we came to this observation only on the basis of the number of days for which the spheres reached a size of 150 microns. So we deleted the phrase: “With that, the ability to form neurospheres and the rate of their growth did not depend on the grade of the primary brain tumor”.
- Figure 1B: please correct labeling of BR1, BR2, MG1 samples in PCA plot; I assume from the text that all samples in the lower left part of the plot are "adherent" and should be labeled as "a", but they are all labeled as "n".
Response: We are very grateful to the reviewer for identifying an error in the notation of the data. We have fixed this in the new version of Figure 1B.
- In line 252, authors claim: "(...) NS formation has the common transcriptional features - gene patterns for all analyzed gliomas" but U87n and U343 cluster with their adherent counterparts and not with BR1n, BR2n, MG1n, which argues against this statement. Did authors refer exclusively to patient derived cell lines? Please, clarify.
Response: Here we once again express our gratitude to the reviewer for the identified flaw in the text. We modified text in the line 252: "Thus, the formation of NS has common transcriptional features and, possibly, common gene patterns for all analyzed gliomas."
- Many of the conclusions reached by the authors rely on the pathway enrichment analyses presented in tables 3 and 4. However, it is not clear whether authors are exclusively employing adjusted p-values as a threshold to consider these pathways significant. Since a high number of pathways are being tested simultaneously, p-values must be adjusted, and adjusted p-values must be used as thresholds of significance. It is also important to clarify whether gene sets composing each transcription factor-associated pathway (for example "SOX2 CHEA") exclusively contain targets that are upregulated by the specific transcription factor or if they also contain genes that are downregulated. This is relevant to interpret what enrichment of upregulated genes in NS vs MN in these pathways means.
Response: We agree with reviewer. In the Table 3 we represented transcription factors controlling gene expression during NS formation and in Table 4 – cellular processes and signaling pathways. Both tables are based on data obtained using Enrich [https://maayanlab.cloud/Enrichr/]. Table 3 - results of up/downregulated gene sets analysis in Enrichr library "ENCODE_and_ChEA_Consensus_TFs_from_ChIP-X". Table 4 - results of gene set enrichment analisys in the Enrichr library "MSigDB_Hallmark_2020".
It should be noted that Enrichr's primary results for Tables 3 and 4 are presented in Supplementary Table S2, which is clearly indicated in the text each time we discuss related results. For example, lines 287-289: "The downregulation of transcription factors MYC and MAX activity is observed in Enrichr data for both specific MN/NS pairs and the corresponding set of overlapping genes (Table 3 and Table S2)". Supplementary Table S2 contains all Enrichr columns, including "P-values" and "Adjusted P-values". In Table 3 and also in Table 4, we present only the 10 most significant Enrichr results, ordered by descending P-value. These details are presented in the headings of the tables - lines 280-282 and 517-519.
It is important to note that in the case where the adjusted p values > 0.05 (greater than the cutoff), we excluded record form the table or indicated this with an asterisk (*).
For example:
- Table 3, column "Overlapped", sub-column "Upregulated", "TCF3 ENCODE*", "TCF3 CHEA*" and "VDR CHEA*";
- Table 4, column "Overlapped", sub-column "Downregulated", ranks 7, 8, 9, 10.
Records marked due to adjusted p values > 0.05 are noted in tables footnote with text: "adjusted p-value > 0.05" (Lines 283 and 522).
On the question of the reviewer "whether gene sets composing each transcription factor-associated pathway (for example "SOX2 CHEA") exclusively contain targets that are upregulated by the specific transcription factor or if they also contain genes that are downregulated".
We should announce the following.
Lists of upregulated genes have no any overlaps with corresponding lists of downregulated genes. Simply because of the algorithm for creating these lists. Thus, for example, BR1 overexpressed genes never overlap with BR1 downregulated genes, and so on for all other cell lines (and, of course, for overlapped gene sets). Thus, upregulated BR1 "SOX2 CHEA" gene set does not contain any of BR1 downregulated genes etc.
- Expression (RNA-Seq counts) of selected genes for all the cell lines analyzed in the different culture conditions are reported in figure 4 and used to substantiate many claims along the paper, such as "(...) for 4 out of 5 analyzed cultures (except for MG1), an increase in SOX2 mRNA level in NS cell cultures was detected" (lines 302, 302). Is SOX2 significantly upregulated when running differential expression analyses by DESeq2 and comparing NS vs MN in all these cell lines? Same question applies for all the genes represented in this figure, please indicate whether they are significantly changed in the different cell lines and culture conditions.
Response: We agree with the reviewer. First. Figure 4 illustrates expression data for a number of differentially expressed genes that we have selected for consideration (like ADAM17 and ARL6). At the same time, Fig. 4 also shows data on the expression of mRNA of transcription factors (TFs) and signal transducer (KRAS). Moreover, changes in mRNA levels of TFs and KRAS often did not meet the criteria for statistically significant unidirectional changes (SOX2, UBTF, MYC, MAX, KRAS).
Practically in Figure 4, we have two different types of data: differentially expressed RNAs; and separately mRNA of TF and KRAS, which are discussed in the article. Both types of transcripts are together alphabetically sorted from left to right in the same figure, and thus this may raise questions.
In order to exclude misunderstanding we changed the background shading of the boxplot headers to white for mRNA of regulatory genes and retained gray shading for genes with differential expression of mRNA. We also added to the Figure 4 caption:
"Plots with gray shaded headers represent data for differentially expressed genes that meet the criteria: DESeq2 padj < 0.05, Log2FoldChange > 0 for upregulated or log2FoldChange < 0 for downregulated transcripts (for all MN/NS pairs in one direction - synchronously "up" or "down"). Plots with white shaded headers represent mRNA levels of TFs and KRAS that do not meet the the criteria for differentially expressed genes (discussed in text)."
With that, we should note that the conclusion for the cited sentences is (lines 304, 305): "Thus, changes in the relative level of SOX2 mRNA were not unidirectional for all analyzed cell cultures under conditions of NS formation."
So, we do not insist on statistical significances of differences between MN and NS SOX2 expression, but rather conclude that changes in SOX2 level were not unidirectional. That practically can be seeing from the data of Figure 4 were SOX2 level in NS of MG1 cell culture is lower than in corresponding MN, while the opposite is true for other cultures. The similar conclusions have been made for others TF discussed: UBTF, NFE2L2, MAX, MYC.
"In our data, the level of UBTF mRNA did not undergo unidirectional changes during the transition from MN to NS (Figure 4)."
"The mRNA level of the NFE2L2, as well as the mRNA level of the SOX2 and UBTF, did not undergo unidirectional changes under the conditions of NS formation (Figure 4)."
"It can be seeing that the level of MYC mRNA, as well as the level of MAX mRNA, changed in different directions during incubation of glioma cells under conditions of NS formation (Figure 4)."
- In lines 308-309, authors state: "It can be assumed that the observed activation of the SOX2 TF is determined to a greater extent by post-translational modifications of the SOX2 protein and/or its interaction with other factors, rather than by the level of its mRNA in glioma cells". The data from which the authors derive this conclusion could also be explained by different expression dynamics of SOX2 in MG1 vs the other cell lines or collection of the samples at a different passage, for example, and no additional experimental data is provided to substantiate this claim. Similar statements are presented along the results section of the paper but are not supported by experimental data, please rephrase to indicate that this is speculation or provide experimental proof.
Response: We agree with the reviewer. We changed the phrase: "It can be proposed that the observed activation of the SOX2 TF is determined to a greater extent by post-translational modifications of the SOX2 protein and/or its interaction with other factors, rather than by the level of its mRNA in glioma cells".
And also line 345. "Therefore, it can be proposed that the observed increase in the expression of UBTF-dependent genes (Figure 4 and 5, Tables 3 and S2) is associated with post-translational activation of the protein and/or its interaction with other factors."
- Lines 502-504: MYC and MAX dimerize to exert their function so it is expectable that that they have shared targets and this does not indicate that they "cooperate in the process of NS formation". Data in figure 4 does not show consistent changes in these two transcription factors in the different cell lines and conditions either.
Response: Thank you for this suggestion. We corrected the phrase “This directly indicates the close cooperation of these TFs in the process of NS formation” to the “This can be explained by the formation of the MYC-MAX heterodimer, which regulates the transcription of target genes”. Indeed, data in figure 4 does not show consistent changes in these two transcription factors in the different cell lines and conditions either. Earlier studies on the differential expression of the MYC, MAX, and RB1 genes in gliomas and glioma cells showed that MYC, MAX, and RB1 are independently regulated in glioma cells (PMID 8286200). There are several related studies that have shown inconsistent data on multidirectional transcription levels of these transcription factors. For example, “Max is stable and ubiquitously expressed, whereas both Myc and Mad family proteins are rapidly degraded and their synthesis is regulated in response to extracellular stimuli” (PMID: 15121849). Recently it was shown that “anaplastic large cell lymphoma had significantly lower MAX expression than peripheral T-cell lymphoma, not otherwise specified, while MYC expression levels were similar between groups both in our study and other data” ….. “Abundant MAX expression generates more MAX-MAX homodimer availability and represses MYC activity through the occupation of DNA binding sites (E-box) of MYC-MAX heterodimer by the homodimer. Decreased MAX protein permits MYC to heterodimerize with MAX instead of MAX-MAX homodimer and to upregulate MYC transcription activity “ (PMID: 32587329). It should also be taken into account that these transcription factors are members of families, each of which can bind to each other or form homodimers, such as the transcription factor MAX. All of the above also explains the absence of a unidirectional change in the levels of these transcription factors in different cell lines and conditions.
- qPCR validation data should be included in the manuscript, represented comparing MN and NS from all the cell lines tested in this study and the potential differences should be statistically tested to validate the changes observed by RNA-Seq. It is also surprising that authors did not validate MYC/MAX, TCF3, STAT3, NFE2L2 or UBTF expression, since these are the TFs the manuscript mostly focuses on.
Response: First, we should note that we used random approach for the validation of RNAseq data. This means that the choice of transcripts for validation should not be based on any hypotheses or assumptions put forward by us in the work, but we should use random transcripts. To diversify the interpretation of the data, it is more important to use random genes, which, in general, are not associated with one or another prerequisite. It's presented in the text lines 608-610: "In order to confirm RNA sequencing data with independent approach we performed qRT-PCR analysis of 8 randomly selected transcripts CXCL1, ERRFI1, NFKBIA, NRP2, PDGFRA, SOX2, TRIB2, and ZEB1."
To highlight this, we have added in "2.7. Real Time RT-PCR Analysis of RNA":
"To confirm the RNAseq results with qRT-PCR, we randomly selected 8 mRNAs: CXCL1, ERRFI1, NFKBIA, NRP2, PDGFRA, SOX2, TRIB2, and ZEB1."
Second. Our data allow to conclude that the levels of key TF's mRNAs SOX2, MYC/MAX, TCF3, STAT3, NFE2L2 or UBTF as well as the level of mRNA KRAS do not reflect the up/down regulation of gene expression in the process of NS formation. Therefore, validation (or non-validation) of the proposed list of transcripts would neither confirm nor invalidate our conclusions.
It should also be noted that validation of RNAseq data (normalized number of sequencing reads) with qRT-PCR data (function of delta-deltaCt) is a case of comparing two sets of independent variables on two different scales. One of the best ways to measure the relationship between two sets of independent experimental data, assuming a linear fit between the data and its normal distribution, is the Pearson correlation coefficient. That is why we used to characterize our RNAseq data vs qRT-PCR results using Pearson correlation coefficient Table S5.
In practice, the data in Table S5 is a representation not only of "comparison of MN and NS of all cell lines tested in this study", but more broadly of comparison of each pair, such as "RNAseq BR1 MN vs qRT-PCR BR1 MN" and "RNAseq MG1 NS vs qRT-PCR MG1 NS" and so on.
The Pearson correlation coefficients in Table S5 (from 0.60 to 0.97) reflect the significance of the linear correlation between RNA sequencing data and qRT-PCR results. Thus it can be concluded (lines 614 -615): "Thus, the high correlation of NGS data with independent qRT-PCR results generally confirms our findings."
- In the discussion, the translational potential of SPRY4, ERRFI1 or RAB31 is discussed, have functional assays downregulating any of these genes in vitro been performed to prove that targeting any of them could be a promising strategy to prevent glioma progression?
Response: Thank you for this suggestion. We did not perform functional assays downregulating any of these genes in vitro, since this was not our goal, they were discovered during a thorough analysis of transcription. Testing the functional role of these factors in this work would make the article redundant.
We added several sentence into the manuscript (Discussion section) to illustrate the functional role of these proteins in the development of glioma: “ERRFI1 has been found to be significant tumor suppressor gene and is frequently deleted, mutated or downregulated in various types of cancer, including glioblastomas [68]. ERRFI1 overexpression has been shown to reduce proliferation in GBM cells by binding EGFR to Syntaxin-8 and targeting internalized EGFR to late endosomes for degradation, while knockdown of ERRFI1 expression resulted in increased tumor invasion [68,79,95]. RAB31 genes products involved in EGFR endocytosis and lysosomal degradation also as ERRFI1. Moreover, RAB31 is on the list of genes with the greatest influence on the development of the highest grade astrocytoma, glioblastoma multiforme. The genes on this list can predict tumor status with 96–100% confidence using logistic regression, cross-validation, and support vector machine analysis [96]. Earlier it was shown the tumor-suppressing role in GBM -derived cell lines of the Spry4 protein which has important functions in many receptor tyrosine kinase -mediated signal transduction cascades [97]. It specifically interferes with MAPK-ERK activation and phospholipase C-induced pathway, affects the PI3K pathway [43]”.
- There are several very relevant studies that were overlooked by the authors (PMID: 19497285, PMID: 34675201 and PMID: 35303420); results presented here should be also discussed in the context of these published findings.
Response: We agree with this and have incorporated your suggestion throughout the manuscript.
All these works are very valuable and we are pleased to pay attention to them. So relevant sentence with these links to these articles (PMID: 19497285, PMID: 34675201 and PMID: 35303420) are included in the discussion section. Véronique G.LeBlanc and colleagues used explants obtained from patients and gliomaspheres in their work. Claire Vinel and co-authors carried out a study on models of tumor-initiating and neural stem cells in glioblastoma. Also in the work of Steven M. Pollard and colleagues, the ability of glioma to give rise to glioma neural stem cell lines (GNS) was demonstrated under adherent cultivation conditions. At the same time, the authors of these articles use other models: there is no direct comparison of the transcriptome changes under the conditions of the neurospheres formation from patient-derived adherent glioma cultures.
We included in the "4. Discussion" direct links to the studies: "There are a number of studies with transcriptome analysis of glioblastoma stem cells, explants and neurospheres [89–92], which demonstrate the relevance of such models for preclinical investigations."
Response to Minor comments:
- Abstract: please, define NGS.
Response: We deleted NGS from the abstract section.
- Abstract: please, explain what "the Enrichr gene sets" means or simply mention the kind of analysis performed (pathway enrichment).
Response: Corrected sentence: "When comparing the transcriptomes of monolayer (ML) and NS cell cultures, we used Enrichr genes sets enrichment analysis to describe transcription factors (TFs) and the pathways involved in the formation of glioma NS."
- Figure 1A: please increase magnification or image resolution, cells are difficult to distinguish.
Response: We corrected images resolution Figure 1A. Inserted MG1a cell image with better distinguishable cells.
- Figure 2: please indicate in figure legend or methods which clustering method was used.
Response: We completed Figure 2 legend with: "The complete agglomeration method for clustering was used."
- Figure 3B: please indicate what green and purple mean in the heatmap.
Response: We deleted green and purple lines as redundant.
Table S3. Sheet name in the Excel file is in Russian, authors may want to translate it to English.
Response: Fixed.
We highlighted (red highlighted text) all changes made when revising the manuscript to make it easier for the Editors to give a prompt decision on manuscript.
We thank the editors for considering our work for publication.
Yours faithfully,
Dr. Dmitry V. Semenov PhD
Associate Professor in Biochemistry
Senior researcher
Laboratory of Biotechnology
Institute of Chemical Biology and Fundamental Medicine SB RAS
(ICHBFM SB RAS)
Russian Federation, 630090, Novosibirsk, Lavrent'ev ave., 8
http://www.niboch.nsc.ru/doku.php/en
tel: 00 7 3833635189
semenov@niboch.nsc.ru

Reviewer 2 Report
The manuscript submitted by Natalia Vasileva et al, entitled “Transcriptome Changes in Glioma Cells Cultivated under Conditions of Neurosphere Formation” reports NGS transcriptomic analysis followed by comparative bioinformatics analysis of glioma cells RNA patterns in the adherent state and in the state of neurospheres. Experimentally, the authors find and select 5 genes, namely SOX2, UBTF, NFE2L2, TCF3 and STAT3. Then, by PCR, they test expression in studied cell lines and cultured glioma tumour cells. Afterwards, using bioinformatics tools and data banks stablish Venn diagrams (elegant and wise), give the list of gene regulated by mentioned SOX2, UBTF, NFE2L2, TCF3 and STAT3 and discuss, according to literature, the importance of these findings.
The manuscript is well cast, well driven, and potentially original, however, short in experimental data supporting specific nuances in GBM tumour biology.
English language revision should be peformed.
Acronims, although trivials (CNS, NGS, and others) should be defined.
Author Response
Dear Reviewers and Editors,
Thank you for giving us the opportunity to submit a revised draft of my manuscript titled Transcriptome Changes in Glioma Cells Cultivated under Conditions of Neurosphere Formation to Cells. We appreciate the time and effort that you and the reviewers have dedicated to providing your valuable feedback on my manuscript. We are grateful to the reviewers for their insightful comments on my paper. We have been able to incorporate changes to reflect most of the suggestions provided by the reviewers.
Here is a point-by-point response to the reviewers’ comments and concerns.
Response to Comments from Reviewer 2
- The manuscript is well cast, well driven, and potentially original, however, short in experimental data supporting specific nuances in GBM tumour biology.
Response. Indeed, this manuscript contains little experimental data confirming certain nuances of the biology of the GBM tumor: the formation of neurospheres, transcriptomic analysis, and validation of transcriptome analysis using RT-PCR. The aim of this study was to determine signaling pathways activated by growth factors that trigger gene expression in order to identify possible targets for diagnostic or therapeutic purposes.
- English language revision should be performed.
Response. We edited the final text using the Language Editing Service (https://www.mdpi.com/authors/english).
- Acronims, although trivials (CNS, NGS, and others) should be defined.
Response. We have added a deciphering of the abbreviation in a special subsection of the text.
We highlighted (red highlighted text) all changes made when revising the manuscript to make it easier for the Editors to give a prompt decision on manuscript.
We thank the editors for considering our work for publication.
Yours faithfully,
Dr. Dmitry V. Semenov PhD
Associate Professor in Biochemistry
Senior researcher
Laboratory of Biotechnology
Institute of Chemical Biology and Fundamental Medicine SB RAS
(ICHBFM SB RAS)
Russian Federation, 630090, Novosibirsk, Lavrent'ev ave., 8
http://www.niboch.nsc.ru/doku.php/en
tel: 00 7 3833635189
semenov@niboch.nsc.ru

Round 2
Reviewer 2 Report
The authors have responded satisfactorily to the reviewer's comments.
Author Response
Response to Reviewer 2 Comments (Round 2)
We thank reviewer for a thorough revision of our manuscript and for your attention to our work, thanks to your comments. The work really looks more complete and harmonious. We hope that the work will be of interest to the readers of the journal and will be useful for understanding the complex processes of glioblastoma oncogenesis.
We highlighted (red highlighted text) all changes made when revising the manuscript to make it easier for the Editors to give a prompt decision on manuscript.
We thank the editors for considering our work for publication.
Yours faithfully,
Dr. Dmitry V. Semenov PhD
Associate Professor in Biochemistry
Senior researcher
Laboratory of Biotechnology
Institute of Chemical Biology and Fundamental Medicine SB RAS
(ICHBFM SB RAS)
Russian Federation, 630090, Novosibirsk, Lavrent'ev ave., 8
http://www.niboch.nsc.ru/doku.php/en
tel: 00 7 3833635189
semenov@niboch.nsc.ru